# Exact and Approximate Algorithms for Polytree Learning

**Juha Harviainen** [* 1]   **Frank Sommer** [* 2]   **Manuel Sorge** [* 3]

## Abstract

Polytrees are a subclass of Bayesian networks that seek to capture the conditional dependencies between a set of $n$ variables as a directed forest and are motivated by their more efficient inference and improved interpretability. Since the problem of learning the best polytree is NP-hard, we study which restrictions make it more tractable by considering for example in-degree bounds, properties of score functions measuring the quality of a polytree, and approximation algorithms. We devise an algorithm that finds the optimal polytree in time $\mathcal{O}\big((2+\epsilon)^n\big)$ for arbitrarily small $\epsilon > 0$ and any constant in-degree bound $k$, improving over the fastest previously known algorithm of time complexity $\mathcal{O}\big(3^n\big)$. We further give polynomial-time algorithms for finding a polytree whose score is within a factor of $k$ from the optimal one for arbitrary scores and a factor of 2 for additive ones. Many of the results are complemented by (nearly) tight lower bounds for either the time complexity or the approximation factors.

## 1. Introduction

Bayesian networks (BNs) are probabilistic graphical models that represent the set of variables and their conditional independencies as a directed acyclic graph (DAG). Widely adopted score-based approaches for learning an optimal structure for the network from data make simplifying assumptions that decompose the posterior probabilities of structures into a product of node-wise *local scores* that depend only on the node and its set of parents in the DAG. Even with these assumptions, however, structure learning

---
[*]Equal contribution  [1]Department of Computer Science, University of Helsinki, Helsinki, Finland [2]Institute of Computer Science, Friedrich-Schiller Universität Jena, Jena, Germany [3]Institute of Logic and Computation, TU Wien, Vienna, Austria. Correspondence to: Juha Harviainen <juha.harviainen@helsinki.fi>, Frank Sommer <frank.sommer@uni-jena.de>, Manuel Sorge <manuel.sorge@ac.tuwien.ac.at>.

*Proceedings of the $43^{rd}$ International Conference on Machine Learning*, Seoul, South Korea. PMLR 306, 2026. Copyright 2026 by the author(s).

remains NP-hard (Chickering, 1995). Further, performing inference is #P-hard (Roth, 1996) in general. Hence, in practice often heuristics are used. One prominent heuristic is K2 (Cooper & Herskovits, 1992): we are given a node ordering, and for each node in that ordering the highest scoring parent set is added while acyclicity is preserved.

These obstacles have motivated studies of constrained classes of BNs that provide a trade-off between the expressiveness and the efficiency of computation. *Polytrees* are BNs whose underlying undirected graph is a forest that are more interpretable and inference takes only polynomial time (Pearl, 1989). In terms of applications, polytrees are used in image-segmentation for microscopy data (Fehri et al., 2019). However, structure learning remains NP-hard for them even when the nodes can have only 2 parents (Dasgupta, 1999), with the fastest known algorithms taking exponential time in the number of nodes (Grüttemeier et al., 2021a) except for the special case of *branchings* where each node is allowed a single parent (Edmonds, 1967). We discover a novel dynamic-programming algorithm for learning the optimal polytree with dynamic programming in time $3^n n^{\mathcal{O}(1)}$ under mild assumptions, matching the time complexity of Grüttemeier et al. (2021a). However, under any constant in-degree bound, we show that the running time can be improved to $\mathcal{O}\big((2+\epsilon)^n\big)$ for arbitrarily small $\epsilon > 0$ by characterizing a subset of dynamic programming states that are sufficient for identifying the optimal structure. We further prove the time complexity to be near-optimal under the set cover conjecture (SCC) of Cygan et al. (2016).

Because of the inherent hardness of learning the structure of an optimal polytree, the rest of the paper focuses in studying the hardness of approximating it, inspired by recent works on approximation algorithms for general BNs (Kundu et al., 2024a; Ziegler, 2008). More precisely, Kundu et al. (2024a) provide so-called FPT-approximations for learning BNs, and Ziegler (2008) studies learning BNs with a maximal parent set size $k$. Interestingly, the algorithm of Ziegler (2008) is a variant of the K2 heuristic: there in each step the highest scoring parent set without violating acyclicity is added (hence no node ordering is used).

We show that without any further assumptions about problem instances, no non-trivial polynomial-time approximation guarantee for polytree learning is possible unless P=NP.

However, if the in-degree of the polytree is again bounded by a constant $k$, then we can get a $(k + 1)$-approximation of the optimal structure, which we also prove to be optimal up to logarithmic factors in $k$ under the Unique Games Conjecture (UG) of Khot (2002). This is somewhat surprising since, as mention above, learning BNs with in-degree bound also admits a factor $k$ approximation (Ziegler, 2008). One could have hoped that the much simpler structure of polytrees allows for a better approximation guarantee. We show, however, that this is not possible. Interestingly, our algorithm is very similar to the one of (Ziegler, 2008): in each step we add the highest scoring parent set, but now we ensure that the underlying graph is acyclic while Ziegler (2008) ensured acyclicity of the directed graph.

Another simplifying assumption motivated by heuristics for BNs (Ganian & Korchemna, 2021; Scanagatta et al., 2016) that we study are *additive* local scores for which each possible arc has a score and the score of a DAG is the sum of the arc-wise scores. Recently Ganian & Korchemna (2026, Theorem 18) showed that Polytree Learning with additive scores can be done in polynomial time. Nonetheless we achieve a 2-approximation of the optimal polytree in that case with the same simple greedy algorithm which always adds the best remaining parent set without violating the polytree constraint. We believe that this result is interesting despite the fact that an optimal solution can be computed in polynomial time since this shows that this simple greedy algorithm can achieve much better approximation guarantees than $k$ if we impose additional constraints on the polytree. In other words, we believe there are scoring functions which are more general than being additive such that this simple greedy algorithm (or a slight adaption of it) yields a constant factor approximation.

Finally, we provide a $2q$-approximation for the variant $\text{PT}_{\text{comp}}$ in which each component is only allowed to have at most $q$ arcs. For a summary of our algorithms and lower bounds, see Table 1.

**Related work.** A closely related optimization problem of minimizing the sum of conditional entropies over the nodes was studied by Dasgupta (1999). Their approximability results, however, do not generalize to the score-based structure learning setting where we instead maximize the sum of local scores.

Alternative structural constraints for learning the optimal structure have been considered especially through the lens of parameterized complexity (see, for example, Grüttemeier & Komusiewicz (2022); Ordyniak & Szeider (2013); Kundu et al. (2024b); Grüttemeier et al. (2021b), and the references therein); also see Ganian (2026) for a survey. In particular for polytrees, Gaspers et al. (2015) study polytrees that are close to a branching, Grüttemeier et al. (2021a)

consider polytrees with bounded in-degree and bounded number of nodes with non-empty parent sets, and Ganian & Korchemna (2021) briefly consider parameterizations for polytrees with additive local scores.

## 2. Preliminaries

Let $D$ be a directed acyclic graph (DAG) over $n$ nodes $V = V(D)$ with a set of arcs $A = A(D)$. The *skeleton* of $D$ is the undirected graph obtained by removing the orientations of its edges. A DAG is a *polytree* if its skeleton is a forest. With mild abuse of terminology, we may also call the arc set $A$ of a polytree if its skeleton is acyclic. A polytree is *connected* if its skeleton is a tree. The set of parents of a node $v \in V$ is denoted by $D_v$.

In the score-based approach to learning BNs, we are given access to a *local score function* $f_v : 2^{V \setminus \{v\}} \to \mathbb{R}$ for each node $v \in V$ that assigns a value for each possible parent set of $v$, typically measuring its quality. The *score of a DAG* is the sum of the local scores, $f(D) := \sum_{v \in V} f_v(D_v)$.

The local score functions need to be described in the input, and without any constraints this would make the input size exponential in $n$. Thus, unless otherwise stated, we adopt the commonly-used *non-zero encoding* of the input, where the input contains a family $\mathcal{F}_v$ of potential parent sets for all $v \in V$ and their associated local scores $f_v(S)$ for all $S \in \mathcal{F}_v$. The unspecified local scores are assumed to be $-\infty$ instead of $0$ despite the nomenclature because, roughly speaking, the scores correspond to the probabilities of the parent sets of which logarithms are taken to improve numerical accuracy. We will further assume that the local scores are normalized in such a way that $f_v(\emptyset) = 0$ for all nodes $v$ by shifting all local scores of each node by an appropriate amount, similarly to Kundu et al. (2024a) and Ziegler (2008). This ensures that the empty graph has score $0$, making the discussion of approximation algorithms more meaningful. Moreover, this shifting does not affect which structure is the optimal one.

Defining $\mathcal{F} := \{(v, S) : v \in V, S \in \mathcal{F}_v\}$ to be the union of all potential parent sets, the size of the input is $|\mathcal{F}|$ up to polynomial factors in $n$. We use $|\mathcal{I}|$ to denote the length of the encoding of the input. We now define our main problem.

Polytree Learning (PT)
*Instance:* Node set $V$, local score functions $f_v$ for $v \in V$.
*Question:* Polytree $D = (V, A)$ with maximum score.

We also study variants of the above problems under various restrictions. A local score function $f_v$ is *additive* if we have that $f_v(S) = \sum_{u \in S} f_v(\{u\})$. When dealing with additive scores, we assume the input to contain the $\mathcal{O}(n^2)$ values $f_v(\{u\})$ for all distinct $v, u \in V$. The score function is

*Table 1.* Summary of our algorithms for running time and approximation factors for polytree learning (PT). The subscript '$\leq$' means that we have an in-degree bound $k$, the superscript '+' means additive local scores, and 'comp' means component size bounded by $q$.

| Problem | Time | Approximation factor | Theorem | Comments |
|---|---|---|---|---|
| PT | $3^n|\mathcal{I}|^{\mathcal{O}(1)}$ | Exact | Theorem 3.1 | Same complexity as Grüttemeier et al. (2021a) |
| PT$_{\leq}$ | $(2+\epsilon)^n|\mathcal{I}|^{\mathcal{O}(1)}$ | Exact | Theorem 3.3 | Any $\epsilon > 0$, near-optimal complexity under SCC |
| PT$_{\leq}$ | $|\mathcal{I}|^{\mathcal{O}(1)}$ | $k+1$ | Theorem 4.1 | Optimal up to logarithmic factors in $k$ under UG |
| PT$_{\leq}^{+}$ | $|\mathcal{I}|^{\mathcal{O}(1)}$ | 2 | Theorem 4.2 | Proven to be in P (Ganian & Korchemna, 2026) |
| PT$_{\text{comp}}$ | $|\mathcal{I}|^{\mathcal{O}(1)}$ | $2q$ | Theorem 4.3 | Optimal up to logarithmic factors in $q$ under UG |

*additive* if all local scores are additive.

When we are guaranteed that the score function is additive, we add the symbol '+' as a superscript to the problem name. Similarly, if we are guaranteed that the largest parent set in the instance with a specified score has size $k$, we add a subscript '$\leq$' to the problem name.

For a constant $c > 1$, an algorithm $\mathcal{A}$ for PT (and its variants) is a *c-approximation algorithm* if it always outputs a polytree $D$ whose score is within a factor $c$ from the optimal polytree $D^{\text{opt}}$, that is, $f(D) \leq f(D^{\text{opt}}) \leq c \cdot f(D)$.

## 3. Exact Algorithms

Gaspers et al. (2015) provided a first exponential time algorithm with running time $n^{n-2} \cdot 2^{n-1}$. This was later improved to $3^n|\mathcal{I}|^{\mathcal{O}(1)}$ (Grüttemeier et al., 2021a). We start by giving an algorithm with time complexity $3^n|\mathcal{I}|^{\mathcal{O}(1)}$ for solving PT, matching the complexity of Grüttemeier et al. (2021a). Afterwards, we modify our approach to yield a $(2+\epsilon)^n|\mathcal{I}|^{\mathcal{O}(1)}$ time algorithm for PT$_{\leq}$ where $\epsilon$ is arbitrary. Finally, we show that our running time for PT$_{\leq}$ is almost tight, assuming standard complexity theory assumptions.

**Theorem 3.1.** PT *can be solved in time* $3^n \cdot |\mathcal{I}|^{\mathcal{O}(1)}$.

*Proof.* Note that PT is easy to reduce to the problem of learning the highest-scoring connected polytree: Add one additional node $u$ whose only possible parent set is the empty set with score 0. For the other nodes $v$, double the number of potential parent sets $S \in \mathcal{F}_v$ by defining a new potential parent set $S \cup \{u\}$ with the same local score as with $S$. For the bounded in-degree variant of the problem, we also increase the in-degree bound by one. Now, any disconnected polytree can be made a connected polytree by connecting each component to $u$ without affecting the score of the structure, and any connected polytree corresponds to a polytree of the original instance by removing $u$.

For subsets of nodes $S, T \subseteq V$ with $T \subseteq S$, let $Q[S, T]$ denote the score of best connected polytree on the set of nodes $S$ such that only the nodes in $T$ can have non-empty parent sets. The optimal score is then found on $Q[V, V]$. Recall that we assumed without the loss of generality that

the scores are normalized such that the empty parent set has score 0, and therefore the score of the polytree corresponding to $Q[S, T]$ equals the sum of the local scores of the nodes in $T$.

**Initialization.** Let $Q[\{v\} \cup D_v, \{v\}] = f_v(D_v)$ for all $v \in N$ and $D_v \in \mathcal{F}_v$, that is, the connected polytree has only one non-empty parent set, which is $D_v$ for $v$. Initialize all remaining entries $Q[S, \{v\}]$ to $-\infty$.

**Dynamic programming.** To compute $Q[S, T]$, we pick one node $v$ in $T$ and guess its parent set $D_v$ in the optimal connected polytree $D$ for $Q[S, T]$. The recurrence used for computing the table is as follows

$$Q[S, T] = \max_{v \in T} \max_{\substack{D_v \in \mathcal{F}_v \\ D_v \subseteq S}}$$

$$\begin{cases} f_v(D_v) + Q[S \setminus D_v, T \setminus \{v\}], \\ \quad \text{if } D_v \cap T = \emptyset \\ f_v(D_v) + Q[S \setminus ((D_v \cup \{v\}) \setminus \{u\}), T \setminus \{v\}], \\ \quad \text{if } D_v \cap T = \{u\} \text{ for some } u \in S, \\ -\infty, \text{ otherwise.} \end{cases}$$

**Correctness.** Consider an optimal connected polytree $D^{\text{opt}}$. Pick an arbitrary root of its skeleton and consider the permutation $\sigma$ of its nodes where $\sigma(i)$ is the $i$-th node of $D^{\text{opt}}$ that a depth-first search (DFS) would visit if it were performed on the skeleton. Let $T_i = \{\sigma(1), \ldots, \sigma(i)\}$ and

$$S_i = T_i \cup D^{\text{opt}}_{\sigma(1)} \cup \cdots \cup D^{\text{opt}}_{\sigma(i)},$$

where $D^{\text{opt}}_{\sigma(i)}$ is the parent set of the node $\sigma(i)$ in the connected polytree $D^{\text{opt}}$. Intuitively, $T_i$ is the set of the first $i$ nodes that the DFS visits, and $S_i$ includes the parents of the nodes in $T_i$ in the connected polytree.

Note that the subgraph of $D^{\text{opt}}$ induced by every $S_i$ is a connected polytree by the properties of DFS. Also recall that the parent set of a node is allowed to be empty in the recurrence. We prove by induction that $Q[S_i, T_i]$ is at least the score of the subgraph of $D^{\text{opt}}$ induced by $S_i$. This clearly holds for the base case, since

$$Q[S_1, T_1] = Q[\{\sigma(1)\} \cup D^{\text{opt}}_{\sigma(1)}, \{\sigma(1)\}] = f_{\sigma(1)}(D^{\text{opt}}_{\sigma(1)}).$$

For $Q[S_i, T_i]$, there are two possible cases depending on if $\sigma(i)$ is in $S_{i-1}$ or not. For conciseness, let $v = \sigma(i)$. If $v \in S_{i-1}$, then we know that all parents of $v$ must be from $V \setminus S_{i-1}$, since $S_{i-1}$ induces a connected polytree on $D^{\text{opt}}$ and $v$ must be a child of some node in $T_{i-1}$. Thus, $D_v^{\text{opt}} \cap S_{i-1} = \emptyset$. Again, it may be that $D_v^{\text{opt}} = \emptyset$. We therefore get that $Q[S_i, T_i]$ is at least

$$f_v(D_v^{\text{opt}}) + Q[S_i \setminus D_v^{\text{opt}}, T_i \setminus \{v\}]$$
$$= f_v(D_v^{\text{opt}}) + Q[S_{i-1}, T_{i-1}].$$

Similarly, if $v \notin S_{i-1}$, then we know that $D_v^{\text{opt}}$ intersects $S_{i-1}$ by exactly one node, say $u$, since otherwise $D^{\text{opt}}$ would not be a connected polytree. By the properties of DFS, we also have that $u \in T_{i-1}$. Again, $Q[S_i, T_i]$ is bounded from above by

$$f_v(D_v^{\text{opt}}) + Q[S_i \setminus ((D_v^{\text{opt}} \cup \{v\}) \setminus \{u\}), T_i \setminus \{v\}]$$
$$= f_v(D_v^{\text{opt}}) + Q[S_{i-1}, T_{i-1}].$$

It follows that $Q[V, V] = Q[S_n, T_n] \geq f(D^{\text{opt}})$.

It remains to argue that the computation paths of dynamic programming resulting with score that is not $-\infty$ all correspond to connected polytrees. We prove again by induction that the value of $Q[S, T]$ corresponds to a connected polytree on the node set $S$. By the initialization, this clearly holds for the base case of sets $S \subseteq V$ and $T = \{v\}$ for some $v \in V$. Assume now that the claim holds for all sets $S$ and proper non-empty subsets $T'$ of $T$.

If $D_v \cap T = \emptyset$, then consider the optimal connected polytree corresponding to the table entry $Q[S \setminus D_v, T \setminus \{v\}]$. We have that $v$ is a node of that tree and none of the nodes $D_v$ are. Therefore it remains a connected polytree after adding the nodes $D_v$ as the parents of $v$.

If $D_v \cap T = \{u\}$ for some $u \in V$, then consider the optimal connected polytree corresponding to

$$Q[S \setminus ((D_v \cup \{v\}) \setminus \{u\}), T \setminus \{v\}].$$

Now, $v$ is not a node of that connected polytree, but adding the parent set $D_v$ to $v$ connects it to the connected polytree. Since $D_v$ intersects $S \setminus ((D_v \cup \{v\}) \setminus \{u\})$ only on one node, the resulting tree is still a connected polytree. Therefore, we have shown that the dynamic programming recurrence considers only connected polytrees and that any connected polytree can be considered.

**Running time.** There are $3^n$ ways of choosing sets $S$ and $T$ with $T \subseteq S \subseteq V$, and computing each dynamic programming state takes $|\mathcal{F}| \cdot n^{\mathcal{O}(1)}$ time. $\square$

The node ordering in Theorem 3.1 was not important. We now show that there is a specific node ordering which yields a helpful property which will be crucial in obtaining a lower running time for $\text{PT}_{\leq}$.

**Lemma 3.2.** *For every connected polytree $D$, there exists a permutation $\sigma$ of nodes such that $|S_i| - |T_i| = \mathcal{O}(k \log n)$ for all $i \in [n]$, where $T_i := \{\sigma(1), \ldots, \sigma(i)\}$ and*

$$S_i := T_i \cup D_{\sigma(1)} \cup \cdots \cup D_{\sigma(i)}.$$

*Proof.* Consider a DFS on the skeleton of $D$ where the DFS always leaves a node towards the smallest unexplored subtree (with ties broken arbitrarily), and let $\sigma$ be the permutation of the nodes describing the order in which DFS visits them. Denote the size of the subtree of $D$ rooted at a node $v$ by $\text{sz}(v)$. Suppose we have run the DFS for an arbitrary number of $i$ time steps, i.e., the search has visited nodes $\sigma(1), \sigma(2), \ldots, \sigma(i)$. We next argue that there are at most $\mathcal{O}(\log n)$ nodes $v \in T_{i-1}$ with $D_v \not\subseteq T_i$. The lemma then follows from that and the in-degree bound.

Observe that if $D_v \not\subseteq T_i$, then there is at least one unexplored subtree rooted at an unexplored neighbor $u$ of the node $v$. Therefore, the DFS is currently exploring the subtree rooted at some other neighbor $w$ of $v$. Consequently, $\text{sz}(v) > \text{sz}(u) + \text{sz}(w) \geq 2 \cdot \text{sz}(w)$, meaning that the subtree rooted at $w$ has fewer than half of the nodes of the subtree rooted at $\text{sz}(v)$. Hence, there can be at most $\mathcal{O}(\log n)$ nodes with $D_v \not\subseteq T_i$.

The bound is tight up to constant factors, since it is achieved by a $k$-ary tree of $n$ nodes. $\square$

Perhaps surprisingly, the previous algorithm can be modified to solve $\text{PT}_{\leq}$ in almost time $\mathcal{O}(2^n)$ by avoiding redundant computation paths in the dynamic programming.

**Theorem 3.3.** $\text{PT}_{\leq}$ *can be solved in time $(2 + \epsilon)^n |\mathcal{I}|^{\mathcal{O}(1)}$ for any $\epsilon > 0$.*

*Proof.* By Lemma 3.2, for every polytree $D$ there exists $\sigma$ such that $|S_i| - |T_i| = \mathcal{O}(k \log n)$ for all $i$. By analogous analysis to Theorem 3.1, we can show that $Q[V, V] \geq f(D^{\text{opt}})$ even if we only consider dynamic programming table entries $Q[S, T]$ with $|S| - |T| = \mathcal{O}(k \log n)$. Therefore, there are only $2^n \cdot n^{\mathcal{O}(k \log n)}$ sets $S$ and $T$ to consider. Further, $\mathcal{F}$ has at most $\mathcal{O}(n^k)$ sets. By adding an arbitrarily small $\epsilon > 0$ to the base of the exponent, all subexponential factors vanish. $\square$

We will now show this complexity to be nearly optimal. In the $k'$-SET COVER problem, one is given a universe of size $n'$ and a family of its subsets, each of size at most $k'$. Then, one should find $t \leq n'$ subsets whose union is the universe. The Set Cover Conjecture (SCC) claims that this problem cannot be solved faster than $\mathcal{O}(2^{(1-\epsilon)n'})$ for any $\epsilon > 0$.

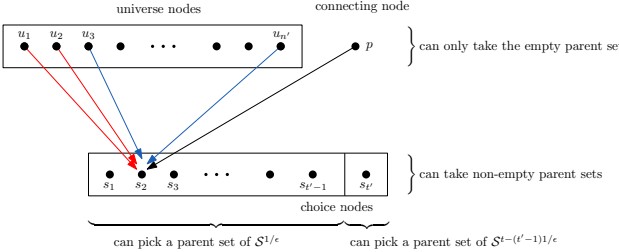

*Figure 1.* A schematic visualization of the reduction provided in Theorem 3.5. Assume that $1/\epsilon = 2$. Then the choice node $s_2$ gets a parent set of $\mathcal{S}^{1/\epsilon} = \mathcal{S}^2$ which consists of the connecting node $p$ and the two sets $\{u_1, u_2\}$ (in red) and $\{u_3, u_{n'}\}$ (in blue) of the $k'$-SET PARTITIONING instance and this gives score 2.

**Conjecture 3.4** (Set Cover Conjecture (Cygan et al., 2016)). *For every $\epsilon > 0$, there is a positive integer $k'$ such that no algorithm can solve $k'$-SET COVER in time $\mathcal{O}(2^{(1-\epsilon)n'})$.*

The $k'$-SET PARTITIONING problem similarly asks if one can find $t$ *disjoint* subsets whose union is the universe. This problem has the same lower bound under SCC, since one can reduce from $k'$-SET COVER by including all $2^{k'}$ subsets of each subset of the universe (Cygan et al., 2016).

We now use the SCC to show that our $(2 + \epsilon)^n |\mathcal{I}|^{\mathcal{O}(1)}$ time algorithm for PT$_\leq$ (Theorem 3.3) is essentially tight.

**Theorem 3.5.** *Under SCC, for any $\epsilon > 0$, there is a constant maximum parent set size $k$ such that no algorithm can solve PT$_\leq$ in time $2^{(1-\epsilon)n}|\mathcal{I}|^{\mathcal{O}(1)}$. Further under SCC, no algorithm can solve PT in time $2^{(1-\epsilon)n}|\mathcal{I}|^{\mathcal{O}(1)}$ for any $\epsilon > 0$.*

*Proof.* Take an arbitrary $k'$-SET PARTITIONING instance with universe $U$ with $n' = |U|$, family of subsets $\mathcal{S} = \{S_1, S_2, \ldots, S_m\}$ of $U$, and an upper bound $t \leq n'$ for the number of disjoint sets we can pick to cover the universe. For contradiction, suppose we can solve PT$_\leq$ in time $\mathcal{O}(2^{(1-\epsilon)n})$ for some fixed $\epsilon > 0$. Without loss of generality, we can assume that $1/\epsilon$ is an integer.

We next define our reduction from $k'$-SET PARTITIONING to PT$_\leq$. For a visualization, we refer to Figure 1. For each element of the universe $u \in U$, create a node with the same label $u$. Then, create $t' := \lceil \epsilon \cdot t \rceil \leq \epsilon \cdot n' + 1$ nodes $s_1, s_2, \ldots, s_{t'}$. Finally, create one additional node $p$. We call these *universe nodes*, *choice nodes*, and the *connecting node*, respectively. In total we have $n = n' + \lceil \epsilon \cdot t \rceil + 1 \leq (1 + \epsilon)n' + 2$ nodes.

For a positive integer $c$, let $\mathcal{S}^c$ be the set

$$\left\{ \{p\} \cup \bigcup_{i \in I} S_i \colon I \subseteq [m], |I| \leq c, S_i \cap S_j = \emptyset \text{ for } i, j \in I \right\}$$

be the family of unions of at most $c$ disjoint sets of $\mathcal{S}$ and $\{p\}$. Let the only potential parent set of universe nodes and the connecting node be the empty set, with score 0.

For choice nodes $s_1, s_2, \ldots, s_{t'-1}$, let the potential parent sets be $\mathcal{S}^{1/\epsilon}$ with the score of a set $S \in \mathcal{S}^{1/\epsilon}$ being $|S|$. Define the potential parent sets similarly for $s_{t'}$ but with $S \in \mathcal{S}^{t-(t'-1)/\epsilon}$ to deal with the remainder. Since the subsets of $U$ are of size at most $k'$, the parent sets of the choice nodes are of size at most $k'/\epsilon$ and their total number is at most $(n')^{k'/\epsilon}$. The parent set size is thus at most a constant for when $\epsilon$ and $k'$ are fixed. The whole construction takes $\mathcal{O}\left((n')^{k'/\epsilon + \mathcal{O}(1)}\right)$ time, which is polynomial if both $\epsilon$ and $k'$ are constants.

We claim that this instance of PT has a solution of score $n'$ if and only if there is a solution to the $k'$-SET PARTITIONING instance. Suppose a solution to the latter exists with sets $S_{i_1}, S_{i_2}, \ldots, S_{i_t}$ for indices $i_1, i_2, \ldots, i_t \in [m]$. Then, construct a polytree where the parents of $s_j$ are

$$\{p\} \cup \bigcup_{l=(j-1)/\epsilon+1}^{\min\{j/\epsilon, t\}} S_{i_l}.$$

That is, the parents of $s_1$ are $p$ and those contained in $S_{i_1}, S_{i_2}, \ldots, S_{i_{1/\epsilon}}$, the parents of $s_2$ are $p$ and those contained in $S_{i_{1/\epsilon+1}}, S_{i_{1/\epsilon+2}}, \ldots, S_{i_{2/\epsilon}}$, and so on. This polytree has a score of $n'$.

For the other direction, assume instead that we have a polytree whose score is at least $n'$. We know that every choice node has to have $p$ as their parent, meaning that the choice nodes cannot share any other parents for the structure to be a polytree. Since the score of the polytree is at least $n'$, each universe node has a single child, which is a choice node. This also implies that the score is $n'$. By the construction of the parent sets of the choice nodes, we can straightforwardly convert the parent sets back into choices of subsets of the $k'$-SET PARTITIONING instance. It follows that the subsets are disjoint and cover the universe.

Under SCC, there is a constant $k'$ such that we cannot solve $k'$-SET PARTITIONING in time $\mathcal{O}(2^{(1-\epsilon^2)n'})$. However, we can build a PT$_\leq$ instance from any such $k'$-SET PARTITIONING instance with $n = (1 + \epsilon)n' + 2$ nodes such that the construction takes polynomial time and has a constant upper bound for the parent set size $k = k'/\epsilon$. We can then solve the instance in time $\mathcal{O}(2^{(1-\epsilon)((1+\epsilon)n'+2)}) = \mathcal{O}(2^{(1-\epsilon^2)n'})$. This violates SCC, resulting in a contradiction. The result follows for PT identically, since we can just use the same local scores in our construction. $\square$

## 4. Approximation Algorithms

In this section, we give our approximation algorithms for bounded in-degree graphs. We start by showing that PT$_\leq$ has a $(k + 1)$-approximation. We then show that a similar algorithm yields a 2-approximation for PT$_\leq^+$. Finally, we consider another restriction that limits the component size.

**Theorem 4.1.** $PT_{\leq}$ *admits a polynomial-time $(k+1)$-approximation algorithm.*

*Proof.* Consider a simple algorithm where we repeatedly add the highest scoring parent set $Q$ for a node $q$ such that (i) a parent set has not yet been chosen for the node $q$ and (ii) adding $Q$ would not violate the acyclicity property of the polytree. Recall that the empty parent set has score 0. Hence, every non-trivial parent set added by this algorithm has score larger than 0. Moreover, it is possible that the algorithm assigns the empty parent set to some nodes. We next prove that this yields a $(k+1)$-approximation.

Let $D^{\mathrm{opt}}$ be the optimal polytree. For convenience, write $V = \{1, 2, \ldots, n\}$ and $f_i := f_i(D_i^{\mathrm{opt}})$ for $i \in V$. Assume without loss of generality that $f_1 \leq f_2 \leq \cdots \leq f_n$. Let $D^i$ be the graph after the $i$-th iteration of our algorithm, that is, the parent sets $Q_1, Q_2, \ldots, Q_i$ have been chosen for the nodes $q_1, q_2, \ldots, q_i$, respectively. We will prove the claim by showing that for the first $\lceil n/(k+1) \rceil$ iterations $i$ we can add a parent set with score at least $f_{(k+1)i+1}$. The result then follows by observing that

$$f(D^{\mathrm{opt}}) := \sum_{i=1}^{n} f_i \leq (k+1) \cdot \sum_{i=1}^{\lceil n/(k+1) \rceil} f_{(k+1)(i-1)+1}$$
$$\leq (k+1) \cdot f(D^n).$$

Observe that there are only two possible reasons for why the parent set $D_j^{\mathrm{opt}}$ of node $j$ could not be added to $D^i$: Firstly, it can be that $j$ already has a fixed (possibly empty) parent set in $D^i$. This can occur only for $i$ nodes $j$. The second case is that $D_j^{\mathrm{opt}} \cup \{j\}$ has at least 2 nodes $u$ and $v$ in some connected component $C$ of $D^i$. In the latter case, say that $u$ and $v$ are connected by a virtual edge from $D_j^{\mathrm{opt}}$. Note then that $C$ can have at most $|C| - 1$ virtual edges from $D_1^{\mathrm{opt}}, D_2^{\mathrm{opt}}, \ldots, D_{ki}^{\mathrm{opt}}$, since otherwise $D^{\mathrm{opt}}$ would contain a cycle. The maximum number of parent sets that cannot be added to $D^i$ for the second reason is therefore bounded from above by the sum $\sum_C |C| - 1$ over the connected components $C$ of $D^i$, which attains its maximum $(k+1)i$ if all sets $Q_{j'} \cup \{j'\}$ are disjoint for all $j' \in [i]$. Therefore, one of the parent sets $D_j^{\mathrm{opt}}$ for $j \in [(k+1)i+1]$ can be added on the $i$-th iteration, and so the added parent set has at least the score $f_{(k+1)i+1}$. $\square$

See Figure 2 for an example where the approximation factor of $k+1$ occurs.

Our second approximation algorithm concerns additive scores with an in-degree bound. Recall that $PT_{\leq}^+$ can be solved in polynomial time Ganian & Korchemna (2026, Theorem 18). Moreover, note that without the in-degree bound, the problem is also solvable in polynomial time (Edmonds, 1967).

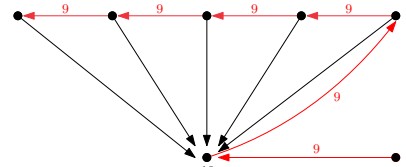

*Figure 2.* An example for a bad local optima where $k = 5$. The polytree $T$ consisting of the black arcs with score 10 is found by the greedy algorithm which always adds the parent set with largest score while preserving acyclicity. The polytree $T^\star$ consisting of the red arcs has a score of 54 and non of the arcs of $T^\star$ can be added to $T$.

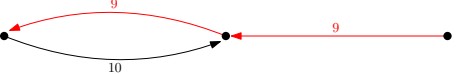

*Figure 3.* An example for a bad local optima where $k = 1$. The polytree $T$ consisting of the black arc with score 10 is found by the greedy algorithm which always adds the edge with largest score while preserving acyclicity and maximum in-degree 1. The polytree $T^\star$ consisting of the red arcs has a score of 18 and non of the arcs of $T^\star$ can be added to $T$.

**Theorem 4.2.** $PT_{\leq}^+$ *admits a polynomial-time 2-approximation algorithm.*

*Proof.* We proceed almost analogously to Theorem 4.1, but instead of adding the whole parent set we greedily add the best edge that does not violate acyclicity or in-degree constraints. This is necessary to avoid bad local optima as shown in Figure 2.

Let $D^{\mathrm{opt}}$ be again the optimal polytree and $D^i$ the graph after the $i$-th iteration of our algorithm. Further, let $e_1, e_2, \ldots, e_{|E(D^{\mathrm{opt}})|}$ be the edges of $D^{\mathrm{opt}}$ such that their scores $f(e_j) := f_v(\{u\})$ with $e_j = (v, u)$ are decreasing in $j$. We argue that there is always an edge with score at least $f(e_{2i+1})$ that can be added.

The two reasons why one of the hightest-scoring edges could not be added are either that the two endpoints of the edge are in the same connected component of $D^i$ or the head of the edge already has $k$ neighbors.

As before, each connected component $C$ of $D^i$ prevents at most $|C| - 1$ edges from being added, and the worst case occurs when all the endpoints of the edges in $D^i$ are distinct, preventing at most $i$ edges from being added.

Further, for each node of $D^i$ with in-degree $k$, the optimal polytree $D^{\mathrm{opt}}$ can contain at most $k$ edges that cannot be added to $D^i$ because of the in-degree constraint. Hence, in-degree constraints prevent at most $i$ edges from being added. Overall, at least on one of the edges $e_1, e_2, \ldots, e_{2i+1}$ can be added to $D^i$, completing the claim. $\square$

We refer to Figure 3 for an example that factor 2 is tight.

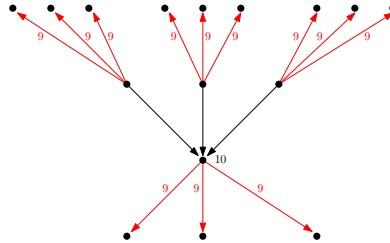

*Figure 4.* An example for a bad local optima for the where $q = 3$. The polytree $T$ consisting of the black arcs with score 10 is found by the greedy algorithm behind Theorem 4.1 which always adds the edge with largest score while preserving acyclicity and maximum component size 3. The polytree $T^\star$ consisting of the red arcs has a score of 108 and non of the arcs of $T^\star$ can be added to $T$.

We now aim to show polynomial time approximation algorithms for further restrictions of PT. More precisely, by $\text{PT}_\text{comp}$ we denote the version in which we require that each connected component of the polytree consists of at most $q$ arcs. Intuitively, one could think that the simple greedy algorithm behind Theorem 4.1 which in each step adds the highest scoring parent set which does not violate acyclicity (and now which also does not violate the size constraint) also achieves a good approximation guarantee for $\text{PT}_\text{comp}$. However, there exist examples (see Figure 4) where this only yields a $\mathcal{O}(q^2)$ approximation. In order to obtain an approximation factor which is linear in $q$ we modify the greedy procedure slightly. For an example where the greedy algorithm behind Theorem 4.1 achieves only a factor $\mathcal{O}(q^2)$ approximation, we refer to Figure 4.

**Theorem 4.3.** $\text{PT}_\text{comp}$ *admits a polynomial-time* $2q$-*approximation algorithm.*

*Proof.* We need the following notation to describe our algorithm: For each parent set $D_v \in \mathcal{F}$ we define $\omega_v(D_v) := f_v(D_v)/|D_v|$, that is, $\omega_v(D_v)$ is a measure for the score contribution of a single arc. For empty parent sets, let $\omega_v(\emptyset) = f_v(\emptyset) = 0$ for all $v \in V$. Moreover, for each $e_{D_v} \in D_v$ we set $\omega_v(e_{D_v}) = \omega_v(D_v)$. Now, consider a simple algorithm where we repeatedly add the highest scoring parent set $W$ according to $(\omega_v(\cdot))_{v \in V}$ for a node $w$ such that (i) a parent set has not yet been chosen for the node $w$, (ii) adding $W$ would not violate the acyclicity property of the polytree, and (iii) adding $W$ does not violate the size bound of any connected component. Note that similar to the proof of Theorem 4.1 since each empty parent set has score 0, every non-trivial parent set added by this algorithm has score larger than 0. Moreover, it is also possible for this algorithm to assign the empty parent set to a node. We next prove that this yields a $2q$-approximation.

By $D_\text{opt}$ we denote the optimal polytree and let $D_\text{opt}^1, \ldots, D_\text{opt}^\ell$ be the connected components of $D_\text{opt}$. Moreover, let $D$ be the polytree computed by our greedy algo-

rithm and let $D^j$ be the polytree after the $j$-th iteration of the greedy algorithm. We say that polytree $D^j$ *touches* a connected component $D_\text{opt}^i$ of $D_\text{opt}$ if any node of $V(D_\text{opt}^i)$ is a head or a tail of an arc of $D^j$. Also, we say component $D_\text{opt}^i$ is *touched* if at least one arc of $D$ touches $D_\text{opt}^i$.

Note that each connected component $D_\text{opt}^i$ of $D_\text{opt}$ with more than a single node is touched: Assume towards a contradiction that $D_\text{opt}^i$ is not touched. By definition, $D_\text{opt}^i$ contains at least 1 arc and thus consists of at least 1 parent set. Now, let $D_v$ be any parent set of $D_\text{opt}^i$. Note that $D_v$ can be added to $D$, a contradiction to the termination condition of our greedy algorithm. Hence, each connected component of $D_\text{opt}$ with multiple nodes is touched, and if an isolated node $v$ is not touched, then also $D$ has an empty parent set for $v$.

For each touched connected component $D_\text{opt}^i$ of $D_\text{opt}$ let $\alpha_i$ be the smallest index such that polytree $D^{\alpha_i}$ touches $D_\text{opt}^i$. Let $D_{v_i}$ be the parent set which is added by the greedy algorithm to obtain $D^{\alpha_i}$ from $D^{\alpha_i - 1}$. Moreover, let $T_w$ be any parent set of $D_\text{opt}^i$. Note that both $D_{v_i}$ and $T_w$ are valid candidate parent sets for the $\alpha_i$-th iteration of the greedy algorithm. Thus, $\omega_{v_i}(D_{v_i}) \geq \omega_w(T_w)$. We let $f(D_\text{opt}^i) := \sum_{x \in V(D_\text{opt}^i)} f_x(T_x)$ where $T_x$ is the parent set of $x$ in $D_\text{opt}^i$ In other words, $f(D_\text{opt}^i)$ is the score of $D_\text{opt}^i$. Observe that $f(D_\text{opt}^i) = \sum_{e_\text{opt}^i \in D_\text{opt}^i} \omega_{p(e_\text{opt}^i)}(e_\text{opt}^i)$, where $p(e_\text{opt}^i)$ is the head of arc $e_\text{opt}^i$. By the above we have $\omega_{v_i}(D_{v_i}) \geq \omega_{p(e_\text{opt}^i)}(e_\text{opt}^i)$ for each $e_\text{opt}^i \in D_\text{opt}^i$. Since by definition each component $D_\text{opt}^i$ has at most $q$ arcs, we have $q \cdot \omega_{v_i}(D_{v_i}) \geq f(D_\text{opt}^i)$. We denote this as the *component property*.

Observe that the score of the optimal solution can be decomposed into the scores of each of its connected components:

$$f(D^\text{opt}) = \sum_{i=1}^\ell f(D_\text{opt}^i) \tag{1}$$

$$\leq \sum_{i=1}^\ell \omega_{v_i}(D^{v_i}) \cdot q = q \cdot \sum_{i=1}^\ell \omega_{v_i}(D_{v_i}) \tag{2}$$

$$\leq q \cdot \sum_{v \in V} (|D_v| + 1) \cdot \omega_v(D_v) \tag{3}$$

$$= q \cdot \sum_{v \in V} (|D_v| + 1) \cdot f_v(D_v)/|D_v| \tag{4}$$

$$\leq 2q \cdot \sum_{v \in V} f_v(D_v) = 2q \cdot f(D). \tag{5}$$

Equation 2 follows by the component property. For Equation 3, observe that each $\omega(D^{v_i})$ can be counted at most $(|D_{v_i}| + 1)$ times: once for each endpoint of an arc in the parent set $D_{v_i}$. Moreover, recall that each component $D_\text{opt}^i$ is touched. Hence, by summing over all nodes and counting each parent set $(|D_{v_i}| + 1)$ times, Equation 3 is true. Equation 4 follows by the definition of $\omega(D_{v_i})$. Finally, Equa-

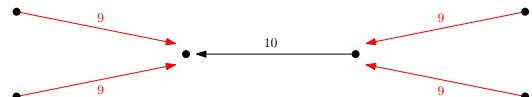

*Figure 5.* An example for a bad local optima where $k = 2$. The polytree $T$ consisting of the black arcs with score 10 is found by the greedy algorithm which always adds the parent set $D_v$ which maximizes $f_v(D_v)/|D_v|$ while preserving acyclicity and the size $k$ restriction for each component. The red numbers represent the number $f_v(D_v)/|D_v|$. The polytree $T^\star$ consisting of the red arcs has a score of 36 and non of the two parent sets of $T^\star$ can be added to $T$.

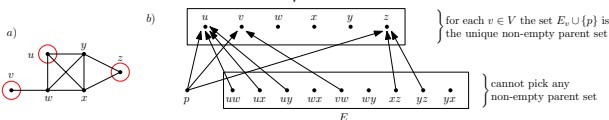

*Figure 6.* Visualization of the reduction of Theorem 5.3. An MAXIMUM INDEPENDENT SET instance is shown in a) where a n independent set of size 3 is shown in red. Moreover, b) shows the constructed PT instance where only the nodes corresponding to the independent set have a non-empty parent set.

tion 5 follows by that fact that $|D_{v_i}|+1/|D_{v_i}|$ is maximized if $|D_{v_i}| = 1$. Hence, we obtain a $2q$ approximation. $\quad\square$

For an example where the approximation factor of $2k$ occurs, we refer to Figure 5.

## 5. Inapproximability

Let us start by recalling some results on hardness of approximation of independent sets, that is, subsets of nodes without edges between them.

**Theorem 5.1** ((Håstad, 1997)). *There is no polynomial-time $\mathcal{O}(n^{1-\epsilon})$-approximation algorithm for* MAXIMUM INDEPENDENT SET *in graphs of $n$ nodes for any $\epsilon > 0$ unless* P = NP.

**Theorem 5.2** ((Austrin et al., 2009)). *There is no polynomial-time $\mathcal{O}(d/\log^2 d)$-approximation algorithm for* MAXIMUM INDEPENDENT SET *in graphs of maximum degree $d$ unless the Unique Games Conjecture is false.*

The following result follows directly by combining the inapproximability result of MAXIMUM INDEPENDENT SET with a reduction of Grüttemeier et al. (2021a, Theorem 2).

**Theorem 5.3.** *There is no polynomial-time $\mathcal{O}(n^{1-\epsilon})$-approximation algorithm for* PT *for any $\epsilon > 0$ unless* P = NP. *Further, there is no polynomial-time $\mathcal{O}(k/\log^2 k)$-approximation algorithm for* PT$_{\leq}$ *unless the Unique Games Conjecture is false.*

*Proof.* We first prove the result for PT. For a visualization of our construction, we refer to Figure 6. Take an arbitrary

instance $G = (V', E)$ for MAXIMUM INDEPENDENT SET. For a node $v \in V'$, let $E_v$ be the set of edges whose endpoint $v$ is. Construct then a PT instance with the node set $V = V' \cup E \cup \{p\}$, where $p$ is a new auxiliary node, and define the local score functions such that the only non-zero scores are $f_v(E_v \cup \{p\}) = 1$ for all $v \in V'$. Call a polytree *reasonable* if there are no nodes with a non-empty parent set with local score 0. We claim that the score of the optimal polytree is precisely the size of the largest independent set, and further, there is a bijection between reasonable polytrees and independent sets.

Let $I$ be an independent set. Consider the reasonable polytree $D$ such that all other parent sets are empty except for all $v \in I$, which have the parent set $E_v \cup \{p\}$. By definition, there are no edges between any two nodes $v, w \in V'$ in $D$. Further, the degree of every node $v \in E$ in $D$ is at most 1, since $I$ is an independent set. Therefore, there are no cycles and the score clearly matches the size of $I$.

Take then any reasonable polytree $D$ and let $I$ be the set of nodes with a non-empty parent set. Suppose that $I$ is not an independent set of $G$. Thus, there are nodes $u, v \in I$ that are connected by an edge $e$ in $G$. Then, however, $D$ has to contain a cycle with edges $u \leftarrow e \rightarrow v \leftarrow p \rightarrow u$, resulting in a contradiction. Since the size of $I$ equals the number of nodes with parents in $D$, its size matches the score of $D$.

Therefore, if we can compute a $c$-approximation of an optimal polytree in polynomial time, we also obtain a $c$-approximation of the maximum-size independent set. To extend the result for PT$_{\leq}$, simply note that the maximum in-degree $k$ in the graph is one greater than the maximum degree $d$ of $G$. A $\mathcal{O}(k/\log^2 k)$-approximation of the optimal polytree would then again yield a $\mathcal{O}(d/\log^2 d)$-approximation of the maximum-size independent set. By Theorem 5.1 and Theorem 5.2, the ability to compute such approximations would imply that P=NP and that the Unique Games Conjecture is false. $\quad\square$

Similar techniques show that the approximation algorithms of Ziegler (2008) for BNs are near-optimal under Unique Games Conjecture by using the reduction of Grüttemeier & Komusiewicz (2022, Theorem 32). Recall the definition of the general problem:

Bayesian network structure learning (BNSL)
*Instance:* Node set $V$, local score functions $f_v$ for $v \in V$.
*Question:* DAG $D = (V, A)$ with maximum score.

**Corollary 5.4 ($\star$).** *There is no polynomial-time $\mathcal{O}(n^{1-\epsilon})$-approximation algorithm for* BNSL *for any $\epsilon > 0$ unless* P = NP. *Further, there is no polynomial-time $\mathcal{O}(k/\log^2 k)$-approximation algorithm for* BNSL$_{\leq}$ *unless the Unique Games Conjecture is false.*

*Proof.* The proof is nearly identical to Theorem 5.3 with the exception that our vertex set $V$ is the same as in the MAXIMUM INDEPENDENT SET instance $G = (V, E)$ and the local scores $f_v(N_G(v)) = 1$, where $N_G$ is the set of neighbors of $v$ in $G$. Reasonable DAGs $D$ cannot have two non-empty parent sets $D_u$ and $D_v$ such that $\{u, v\} \in E$, since then there would be a cycle $u \leftarrow v \leftarrow u$. $\square$

Finally, we note that a slight modification suffices also for $\text{PT}_{\text{comp}}$.

**Corollary 5.5 (★).** *There is no polynomial-time $\mathcal{O}(q/\log^2 q)$-approximation algorithm for $\text{PT}_{\text{comp}}$ unless the Unique Games Conjecture is false.*

*Proof.* The proof is again nearly identical to Theorem 5.3. We reduce from a MAXIMUM INDEPENDENT SET instance $G = (V', E)$ with maximum degree $d$, and let $q = d + 1$ and $V = V' \cup E$. Define $E_v$ again as the set of edges whose endpoint $v$ is, but and while $|E_v| < d$, add dummy nodes to the graph and to $E_v$. Then, define local scores $f_v(E_v) = 1$ for all $v \in V'$. Any reasonable polytree $D$ cannot have two non-empty parent sets $D_u$ and $D_v$ such that $\{u, v\} \in E$, since each reasonable parent set has size $q - 1$, but $u$ and $v$ are also in the same component, making its size exceed $q$. $\square$

## 6. Conclusion

We provided exponential-time algorithm for PT and $\text{PT}_\leq$ with running times $3^n |\mathcal{I}|^{\mathcal{O}(1)}$ and $(2 + \epsilon)^n |\mathcal{I}|^{\mathcal{O}(1)}$ for an arbitrary $\epsilon$, respectively. We showed that the latter result is almost tight under SCC. We then initiated the study of approximation algorithms for polytrees. We showed that $\text{PT}_\leq$ and $\text{PT}_\leq^+$ have polynomial-time approximation algorithms with factors $(k + 1)$ and 2, respectively. We showed that under UG the former result is almost tight.

An interesting open question is whether our 2-approximation for $\text{PT}_\leq^+$ can be significantly improved. Moreover, it is interesting to see whether the ideas of the FPT-approximation of Kundu et al. (2024a) can be transferred to polytrees. Also, it is open whether an algorithm faster than $3^n \cdot |\mathcal{I}|^{\mathcal{O}(1)}$ for PT is possible.

Currently, learning BNs and polytrees is quite well understood. One the one hand, BNs provide the most general structure (unbounded treewidth) but their inference is NP-hard. On the other hand, inference for polytrees can be done in polynomial time since they have treewidth 1. Hence, it is natural to study a generalization of polytrees which allows more structures but still allows for an efficient inference task. One option is to limit the treewidth tw of the learned network by a constant. Recall that tw = 1 corresponds to polytrees. It is interesting to develop exact and approximation algorithm for these networks as well.

## Acknowledgements

*Juha Harviainen:* Co-funded by the Research Council of Finland, Grant 351156. Co-funded by the European Union (ERC, SCALEBIO, 101169716). Views and opinions expressed are however those of the author(s) only and do not necessarily reflect those of the European Union or the European Research Council. Neither the European Union nor the granting authority can be held responsible for them.

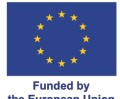 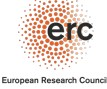

*Frank Sommer:* Supported by the Alexander von Humboldt Foundation and partially by the Carl Zeiss Foundation, Germany, within the project "Interactive Inference".

We also gratefully acknowledge support by the TU Wien International Office for hosting Juha Harvianen in Vienna.

## Impact Statement

This paper presents work whose goal is to advance the field of Machine Learning. There are many potential societal consequences of our work, none which we feel must be specifically highlighted here.

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
