# OpenReview forum: "Exact and Approximate Algorithms for Polytree Learning"
_ICML.cc/2026/Conference — ICML 2026 regular_

### Official Review · Reviewer_fi4j · 2026-03-08

**Soundness:** 4
**Presentation:** 4
**Significance:** 3
**Originality:** 3
**Overall Recommendation:** 5
**Confidence:** 4

**Summary:**

This paper is concerned with obtaining exact and approximate algorithms for polytree learning. A polytree learning instance over $n$ nodes comprises of a node set $V$, together with a family $\mathcal{F}_v$ of sets of potential parent nodes for every vertex $v$, along with a "local score" assigned to each such potential set. The task of the learner is to construct a directed edge set over the vertices such that the realized polytree has maximum score, where the score is the sum of the local scores. Let $\mathcal I$ denote the polytree learning instance, and $|\mathcal I|$ denote size of the input representation. The main results in the paper are as follows:
1) A Dynamic Programming algorithm for exactly computing the optimal polytree, in time $3^n |\mathcal I|^{O(1)}$ --- this matches the running time of a previous algorithm by Grüttemeier et al. 2021.
2) For any constant in-degree bound of $k$ on the target polytree, an improved running time bound of $(2+\epsilon)^n |\mathcal I|^{O(1)}$, and evidence that doing any better is hard under the set cover conjecture.
3) A polynomial time approximation algorithm that runs in time $|\mathcal I|^{O(1)}$, and approximates the maximum score to a factor $k+1$, when the underlying polytree has in-degree at most $k$. This result is also tight under the Unique Games Conjecture.
4) The previous result improves to $2$ approximation if the score functions are "additive".
5) An approximation factor of $2q$ if every connected component in the target polytree has at most $q$ edges.

**Compliance With Llm Reviewing Policy:**

Affirmed.

**Final Justification:**

The author rebuttal suitably addresses my questions, and I maintain my positive evaluation of this paper

**Key Questions For Authors:**

In the induction base case of the Correctness proof of Theorem 3.1, the authors say that $Q[S_i, T_i]$ is at least the score of the subgraph of $D^{opt}$ induced by $S_i$. Shouldn't this instead be "subgraph of $D^{opt}$ induced by $T_i$"? because for $i=1$, the subgraph of $D^{opt}$ induced by $S_1$ could have all the parents of $\sigma(1)$, and so the score of the subgraph of $D^{opt}$ induced by $S_1$ is not just $f_{\sigma(1)}(D^{opt}_{\sigma(1)})$.

**Limitations:**

yes

**Strengths And Weaknesses:**

### Strengths:
The results derived are comprehensive and conceptually clean. The derived algorithms are simple to understand as well. The running time bounds derived are complemented with near-matching lower bounds. In all, this work considerably extends our knowledge about polytree learning algorithms, using simple and elegant analyses.

### Weaknesses:

It is my view that the paper tells a concise and compelling story, with a strong suite of results, and as such, no glaring weaknesses jumped out for me.

### Minor Typos:
Line 143: It should be $Q[V,V]$ instead of $Q[N,N]$.\
Before Conjecture 3.4, should it be $O$ instead of $O^*$?

---

> ### Author Rebuttal · Authors · 2026-03-30
>
> Thank you for the review and pointing out the typos that we will fix!
>
> About your question, we exploit the assumption mentioned in the preliminaries that the scores are shifted in such a way that an empty parent set always has a local score 0. The score of the subgraph of $D^{opt}$ induced by $S_1$ then has score $f_{sigma(1)}(D_{\sigma(1)}^{opt})$, since all the other parent sets in the induced subgraph are empty. We will make this more explicit in the proof and remark that shifting does not affect which structure is the optimal one.

---

> > ### Author Rebuttal · Reviewer_fi4j · 2026-04-01
> >
> > The author's response addresses the question I had, and I have no further concerns

---

### Official Review · Reviewer_FMmN · 2026-03-10

**Soundness:** 2
**Presentation:** 2
**Significance:** 2
**Originality:** 2
**Overall Recommendation:** 5
**Confidence:** 4

**Summary:**

This paper studies the problem of learning optimal polytrees in Bayesian networks under the score-based structure learning framework. Polytrees are a restricted class of Bayesian networks whose underlying undirected structure forms a forest, which enables efficient probabilistic inference while retaining a certain level of expressiveness.
The authors investigate both exact and approximate algorithms for the polytree learning (PT) problem. The paper presents an exponential-time exact algorithm for learning optimal polytrees and improves the running time for the bounded in-degree case from previously known algorithms. Specifically, the authors propose a dynamic programming algorithm with running time for general instances and show that, under a bounded in-degree assumption, the running time can be improved. In addition, the paper studies approximation algorithms for several variants of the problem, including bounded in-degree polytrees, additive score functions, and bounded component sizes. The authors provide polynomial-time approximation algorithms with factors depending on the assumptions. The paper also presents several hardness and inapproximability results to complement these algorithmic guarantees.
Overall, the paper addresses the algorithmic complexity of learning polytrees and provides theoretical results on exact algorithms, approximation algorithms, and lower bounds for this problem.

**Compliance With Llm Reviewing Policy:**

Affirmed.

**Final Justification:**

The authors’ rebuttal satisfactorily addressed my main concerns, leading me to update my overall recommendation to Accept.

**Key Questions For Authors:**

1. While the paper is motivated by Bayesian network structure learning, the contributions are mainly algorithmic and complexity-theoretic. Could the authors clarify the practical machine learning implications of these results?

2. The paper focuses entirely on theoretical analysis and does not include empirical evaluation. Have the authors considered implementing the proposed algorithms and evaluating their performance in practical structure learning scenarios? Such experiments could help demonstrate the practical value of the proposed methods.

3. Many existing approaches for Bayesian network structure learning rely on heuristic search or approximate optimization techniques. How do the proposed algorithms compare with such approaches in terms of scalability and practical usability?

**Limitations:**

The primary limitation of this work is that the contributions are largely theoretical and focus on algorithmic complexity rather than machine learning methodology. While the analysis of exact and approximation algorithms for polytree learning is interesting from an algorithmic perspective, the manuscript does not clearly demonstrate its impact on practical machine learning tasks.
Additionally, the lack of empirical evaluation makes it difficult to assess whether the proposed algorithms are useful in realistic Bayesian network learning settings. Since Bayesian network structure learning often involves large datasets and complex models, experimental validation would be important for establishing the practical significance of the work.
Overall, although the paper provides technically sound theoretical results, the connection to core machine learning research and practical learning applications appears limited, which weakens its suitability for ICML.

**Strengths And Weaknesses:**

[Strengths]
One strength of the paper is that it studies the classical optimization problem of polytree learning within the Bayesian network structure learning framework. The paper provides several theoretical results regarding the computational complexity of the problem and proposes improved exact algorithms under certain structural assumptions such as bounded in-degree. The analysis of approximation guarantees and hardness results is technically sound and contributes to the algorithmic understanding of this problem.
The paper also provides a systematic study of several variants of the polytree learning problem and offers both upper and lower bounds for the achievable approximation ratios under different assumptions.

[Weaknesses]
Despite these theoretical contributions, the relevance of the work to the broader machine learning community is somewhat limited. The main results focus on algorithmic complexity analysis and approximation guarantees for a specific combinatorial optimization problem. While the problem is related to Bayesian network structure learning, the paper does not provide new learning methods, modeling approaches, or empirical evaluations that demonstrate practical impact in machine learning settings.
Furthermore, the manuscript primarily develops algorithmic and complexity-theoretic results rather than machine learning methodology. As a result, the contribution appears closer to theoretical computer science or combinatorial optimization than to core machine learning research typically presented at ICML.
Another limitation is the absence of empirical evaluation. The paper does not include experiments demonstrating how the proposed algorithms perform in practice, for example on real or synthetic datasets for Bayesian network structure learning. Without such evidence, it is difficult to assess the practical relevance of the proposed algorithms for machine learning applications.

---

> ### Author Rebuttal · Authors · 2026-03-30
>
> Thank you for the review.
>
> If we understand correctly, your main concern is that the methods that we introduce might not been empirically evaluated yet. Here is why our results still have direct and indirect implications for practice:
>
> For direct implications: We study the straightforward greedy algorithm that iteratively picks the best-scoring parent set and determine its approximation guarantee precisely under standard theoretical assumptions. This algorithm is for instance implemented here: https://proceedings.neurips.cc/paper/2015/hash/2b38c2df6a49b97f706ec9148ce48d86-Abstract.html and similar greedy algorithms are implemented in standard libraries for Bayesian network learning as subroutines such as in Weka's K2 or Gobnilp.
>
> For indirect implications: The theoretical hardness of the problem is highly likely because the reductions produce hard instances that don't reflect practical instances. We push into the direction of making theory more relevant to practice by studying special cases that either already directly reflect practical settings (moderate parent set sizes) or may point the way towards finding the right formulation of the structure in practical instances (somewhere between arbitrary and additive scores, which we study). Once we have a formulation of practical structure that enables efficient algorithms with good performance guarantees, this will enable better implementations than the heuristics that are currently prevalent.
>
> We hope this answers your questions Q1-3. For further details, see also our response to reviewer DaEc.

---

> > ### Author Rebuttal · Reviewer_FMmN · 2026-04-02
> >
> > The authors’ response satisfactorily addressed my questions, and I have no further concerns.

---

### Official Review · Reviewer_HcAS · 2026-03-12

**Soundness:** 3
**Presentation:** 3
**Significance:** 2
**Originality:** 2
**Overall Recommendation:** 4
**Confidence:** 4

**Summary:**

The paper studies the problem of constructing high score polytree, a subclass of bayesian network where the target network is restricted to be a forest. The authors present an exact exponential algorithm based on dynamic programming that matches the state of the art for the general case. This algorithm is then used as a basis for a more efficient (still exponential) algorithm for the bonded in-degree variant of the problem. The algorithm is then proved to match a lower bound derived by a reduction from Set Cover (assuming the set cover conjecture of Cygan et al.). A simple k-approximation algorithm is also shown for the k-bounded in-degre variant. Stronger constant approximation guarantees are then shown under the additional assumption that the scores are additive. Inapproximability results are given to complement the positive results.

**Compliance With Llm Reviewing Policy:**

Affirmed.

**Key Questions For Authors:**

N/A

**Limitations:**

yes

**Strengths And Weaknesses:**

All the claims are soundly supported by proofs. The presentation is very clear.
The significance of the results is mostly theoretical. It is nice that most algorithmic results are complemented by (almost) matching lower bounds, based on standard complexity conjectures. About originality, both the reductions and the algorithmic tool employed are not surprising but this helps the clarity of the presentation.

---

> ### Author Rebuttal · Authors · 2026-03-30
>
> Thanks for your review!

---

> > ### Author Rebuttal · Reviewer_HcAS · 2026-04-02
> >
> > "de facto" I did not have real concerns. I have also read the authors rebuttals to the other reviews, and I am confident with my evaluation

---

### Official Review · Reviewer_DaEc · 2026-03-12

**Soundness:** 4
**Presentation:** 3
**Significance:** 3
**Originality:** 3
**Overall Recommendation:** 5
**Confidence:** 3

**Summary:**

Polytrees are special forms of Bayesian networks for which several
inference problems can be solved efficiently. The paper considers the
problem of constructing optimal polytrees from data. The problem is known
to be NP-hard. The paper presents a faster exponential algorithms when
the number of candidate parents for each node is bounded by a constant.
Some approximation algorithms are also presented for special cases
and complexity results are presented to show that these are nearly
the best possible under some known complexity theoretic hypotheses.

**Compliance With Llm Reviewing Policy:**

Affirmed.

**Final Justification:**

I raised my score for this paper after going through all the reviews and the corresponding rebuttals.

**Key Questions For Authors:**

(a) You cite (Pearl, 1989) to point out that the inference problems can be
solved efficiently for polytrees. Are there references that point out
that some of the special cases of the polytree learning problem considered
in this paper arise in practice?

(b) The algorithms presented in some of the proofs (e.g., the first
paragraphs of the proofs of Theorems 4.1 and 4.3) seem too short
to get a good understanding. For example, in the algorithm sketched
in the proof of Theorem 4.1, would it be possible that we end up
without a parent set for a node q (since such a set satisfying
conditions (i) and (ii) can't be found)? It will be better to make
the descriptions  a bit longer to make them easier to understand.

**Limitations:**

Yes

**Strengths And Weaknesses:**

Strengths:

(a) The problem of constructing polytrees from data has been studied in a
number of papers in the AI literature. The paper makes novel theoretical
contributions.

(b) The complexity results connecting the problem to other well known
combinatorial problems (such as Maximum Independent Set) are interesting.

(c) Overall, the writing style is good and provides a good coverage of
prior work on the topic.

Weaknesses:

(a) The contributions are theoretical. It is not clear whether the
special cases considered in the paper are of practical interest.

(b) The paper would have been stronger with experimental results to
illustrate the practical performance of the algorithms. Without such results,
it is not clear whether this work will appeal to the ICML community.

---

> ### Author Rebuttal · Authors · 2026-03-30
>
> Thank you for the review! Please see below for our responses to the Reviewer’s concerns:
>
> *Considered special cases*
>
> The main constraints considered in the paper are the in-degree bound and additive scores. The in-degree bound is very prominent in the literature on Bayesian networks and in practice, since with an in-degree bound $k$, the algorithms need to consider only $O(|V|^k)$ parent sets. This does not typically affect the quality of the learnt structure significantly, since the local scores tend to be maximized for small parent sets. Employing just small parent sets not only speeds up the computations and simplifies the structure but also makes the conditional probability tables of the network smaller. Finally, many algorithms utilize precomputed local scores rather than computing them on the fly, and without the in-degree bound the size of the score file would grow exponentially in the number of vertices.
>
> Typically in structure learning literature, one makes zero assumptions about the local scores and just treats them as black-box functions. This lack of structure then allows one to construct arbitrary—likely unrealistic—instances for proving many things to be NP-hard. Additive local scores were inspired in the literature by heuristics for structure learning and can be exploited by algorithms because of how “structured” they are (see, e.g., Ganian and Korchemna, 2021, and the references therein).
>
> We wish to note that while the paper was under review, Ganian and Korchemna (2021) uploaded an updated version of their manuscript on arXiv, which showed that learning with additive scores and in-degree bounds is solvable in polynomial time contrary to what was previously assumed. Nevertheless, we think that our approximation algorithms for additive scores can serve as a starting point for designing algorithms for score functions with weaker assumptions, e.g., subadditive or submodular scores, which might be closer to the behavior that the score functions typically exhibit.
>
> *Adding details to proofs*
>
> We thank the Reviewer for bringing this to our attention, and will clarify these proofs accordingly. To answer the specific question of the Reviewer, it is indeed possible that some node ends up with an empty parent set, but these are also valid sets to be added as they satisfy properties (i) and (ii). In fact, the algorithm would achieve the approximation guarantee even if it picked empty parent sets for all remaining nodes after the first $n/(k+1)$ iterations; the core of the proof is that the first $n/(k+1)$ chosen parent sets already have a relatively high score.
>
> *Theoretical focus of the paper*
>
> The main focus of the paper was in algorithms with performance guarantees for the running time and the quality of the solution, which are features that many heuristics lack—especially the latter one. Our hope is that our techniques can be refined in the future to design practical algorithms for structure learning with accuracy guarantees that then perform decently in the worst case and in practice even better. However, designing meaningful experiments for the performance appears difficult due to structure learning being NP-hard, and the optimum would be needed for assessing the approximation ratios.
>
> Finally, we wish to note that a significant portion of recent literature on Bayesian networks is purely theoretical, even among top ML venues (see, e.g. Ganian and Korchemna, 2021; Grüttemeier et al., 2021a), so we do not consider the theoretical focus of the paper a weakness in itself.

---

> > ### Author Rebuttal · Reviewer_DaEc · 2026-04-03
> >
> > I thank the author(s) for a careful rebuttal that addresses all my concerns.  I have also gone through the other reviews and the corresponding rebuttals. I have raised my score from "weak accept" to "accept".

---

### Decision · Program_Chairs · 2026-04-30

**Decision:**

Accept (regular)

**Comment:**

The paper studies a well-known subclass of Bayes nets: polytrees. The authors provide a $O((2+\epsilon)^n)$ algorithm for $n$ variables, improving on the previously-known $O(3^n)$ algorithm. Polynomial time approximation algorithms are also provided. Finally, negative results are provided under $P \neq NP$ or unique games conjecture.

The reviewers unanimously agree to accept this paper. For a camera-ready version, please take into the account the comments from all reviewers. Implementing the algorithms and/or using experimental evidence from prior literature might help better appeal to the ML community.